# TCAV: Relative concept importance testing with Linear Concept Activation Vectors

## Abstract

Neural networks commonly offer high utility but remain difficult to interpret. Developing methods to explain their decisions is challenging due to their large size, complex structure, and inscrutable internal representations. This work argues that the language of explanations should be expanded from that of input features (e.g., assigning importance weightings to pixels) to include that of higher-level, human-friendly concepts. For example, an understandable explanation of why an image classifier outputs the label "zebra" would ideally relate to concepts such as "stripes" rather than a set of particular pixel values. This paper introduces the "concept activation vector" (CAV) which allows quantitative analysis of a concept's relative importance to classification, with a user-provided set of input data examples defining the concept. CAVs may be easily used by non-experts, who need only provide examples, and with CAVs the high-dimensional structure of neural networks turns into an aid to interpretation, rather than an obstacle. Using the domain of image classification as a testing ground, we describe how CAVs may be used to test hypotheses about classifiers and also generate insights into the deficiencies and correlations in training data. CAVs also provide us a directed approach to choose the combinations of neurons to visualize with the DeepDream technique, which traditionally has chosen neurons or linear combinations of neurons at random to visualize.

## 1 Introduction

Neural networks (NNs) are capable of impressively good performance, yet understanding and interpreting their behavior remains a significant challenge. Solving this challenge is an important problem for several reasons. For example, explaining a system's behavior may be necessary to establish acceptability and see adoption for critical applications, such as those in the medical domain. For scientists and engineers, any greater understanding of how neural networks function is appreciated, since it may lead to better models and help with debugging (30; 19).

Recent work suggests that linear combinations of neurons may encode meaningful, insightful information (2; 19; 27). However, we lack methods to 1) identify which linear combinations (if any) relate to a given concept, and 2) how these can aid in our quantitative understanding of concepts and classification decisions. For example, we may hypothesize that an image model that successfully classifies zebras may naturally encode concepts for 'stripe' and 'animal', somewhere in its internal representations, using a linear combination of neurons. How can we formalize this notion, and test such a hypothesis?

Neural networks build internal representations that are far richer than the input features or output classes explicit in their training data. Unfortunately, many machine learning interpretation methods provide results only in terms of input features. For example, the learned coefficients in linear classifiers or logistic regression can be interpreted as each feature's classification importance. Similar first-order importance measures for neural networks often use first derivatives as a proxy for input feature importance, as is done for pixel importance in saliency maps (8; 22).

It is critical that model understanding and interpretation not be limited to only the concepts explicit in training data. This can be seen by considering classification fairness—an increasingly relevant, difficult problem where interpretability can be useful—and noting that no input features may identify discriminated-against groups. For example, the Inception model (26) has an output class for 'doctor'

but no input features identifying the concepts of 'man' or 'woman' in a way that would allow existing interpretability approaches to quantify gender bias in classification.

This work introduces the method of concept activation vectors (CAV) for the following purposes. First, CAV can be used to identify linear combinations of neurons in a layer of a model that correspond to given semantic concepts, even for new, user-provided concepts not explicit in the model's training data. Second, CAV provides quantitative measures of *the relative importance of user-provided concepts*, which allows for hypothesis testing of the relationship between given concepts and the model's predictions.

Testing with CAV (TCAV) is designed with the following desiderata in mind.

1. **accessibility**: Requires little to no user expertise in machine learning.
2. **customization**: Adapt to any concept of interest (e.g., gender) on the fly without pre-listing a set of concepts before training.
3. **plug-in readiness**: Work without retraining or modifying the model.
4. **quantification**: Provide quantitative explanation that are tied to human-relatable concept, and not input features.

One of key ideas for TCAV is that we can test the *relative importance* between small set of concepts, rather than ranking the importance of all possible features/concepts. For example, we can gain insights by testing whether the concept of gender was used *more* than the 'wearing scrubs' concept for the classification of doctor. We can also test whether or not a given concept was relevant to the classification of a certain class. Similar forms of sparsity (i.e., only considering a few concepts at a time) are used in many existing interpretable models (12; 7; 28; 31; 29; 4). Note that interpretability does not mean understanding the entire network's behavior on every feature/concept of the input (6). Such a goal may not be achievable, particularly for ML models with super-human performance (23).

TCAV satisfies these desiderata—accessibility, customization, plug-in readiness and quantification —it enables quantitative relative importance testing for non-ML experts, for user-provided concepts without retraining or modifying the network. Users express their concepts of interest using examples—a set of data points exemplifying the concept. For example, if gender is the concept of interest, users can collect pictures of women. The use of examples has been shown to be powerful medium for communication between machine learning (ML) models and non-expert users (16; 12; 13). Cognitive studies on experts also support this approach (e.g., experts think in terms of examples (15)).

The structure of this paper is as follows: Section 2 relates this work to existing interpretability methods. Section 3 explains the details of the TCAV method. In Section 4, we show 1) how this framework can be used to identify semantically meaningful directions in a layer and 2) the relative importance testing results that measure the relevance of concepts of interest to the classification output by the network.

## 2 RELATED WORK

In this section, we provide a brief overview of existing related interpretability methods and their relation to our desiderata. We also discuss the need and the challenges of desiderata 3): plug-in readiness.

### 2.1 SALIENCY MAP METHODS

One of the most popular approaches in interpreting NN is saliency methods (24; 22; 25; 8; 5). These techniques seek to identify regions of the input most relevant to the classification output by the network. Qualitatively, these methods often successfully label regions of the input which seem semantically relevant to the classification.

Unfortunately, these methods do not satisfy our desiderata of 2) customization and 4) quantification. The lack customization is clear, as the user has no control over what concepts of interest these maps pick up on. Regarding quantification, there is no way to meaningfully quantify and interpret the brightness of various regions in these maps. As a hypothetical example, consider two saliency maps

of two different cat pictures, where the brightness of the ears of the two cats differ. It is unclear both how to quantify 'brightness' and second what kind of actionable insights this level of brightness gives.

Recent work has also demonstrated that the saliency map these methods produced may be very sensitive to completely uninteresting properties of the data distribution. In particular, (14) showed that simply applying a mean shift to the dataset may cause some saliency methods to result in significant changes in the given explanation. (9) also showed that saliency methods can be easily tricked.

## 2.2 DeepDream, neuron level investigation methods

There are techniques, such as DeepDream, which can be used to visualize patterns that maximally activates each neuron of a neural network. The technique starts from an image of random noise and iteratively modifies the image in order to maximally activate a neuron or a linear combination of neurons of interest (17; 18). This technique has offered some insights into the information encoded in a neuron's activation. This technique also has opened up opportunities for AI-aided art (17).

However, the DeepDream method does not satisfy our desiderata 1) accessibility, 2) customization, and 4) quantification. It does not satisfy 1) because in order to apply it a user must first understand what a neuron is, and second be familiar enough with the the internal architecture in order to choose which neurons to visualize. It does not satisfy 2) because no current method exists to find which neurons correspond to semantically meaningful concepts such as gender, and it is unclear whether or not such a neuron even exists. It does not satisfy 4) because we do not understand these pictures and there is currently no method to quantify how these pictures relate to the output of the network. This method again does not provide actionable insight.

As we show later, DeepDream may be combined with TCAV in order to identify and visualize interesting directions in a layer. Prior work on DeepDream has typically chosen neurons or linear combinations of neurons at random to visualize. Note that a user of TCAV does need to pick a layer in the network for which to apply TCAV to. However, if only the final prediction is concerned, the last layer can be used by default.

## 2.3 Why we need desiderata 3 - plug-in readiness

To achieve interpretability, we have two options: (1) restrict ourselves to inherently interpretable models or (2) post-process our models in way that yields insights.

Users may choose option (1) as there are a number of methods to build inherently interpretable models (12; 7; 28; 31; 29; 4). If the users are willing and are able to adapt these models, then this is the gold standard. Although building inherently interpretable machine learning models may be possible for some domains, doing so may result in decreased performance. Furthermore, changing the model may be costly for users who already have a model implemented.

A method that can be applied without retraining or modifying the network could be instantly used to interpret existing models. Increasingly, attention is turning to the importance of providing explanations for ML decisions (for one example, consider (10)). As a result, there is a growing need for interpretation techniques that can be applied "out of the box," that is, without rebuilding or retraining existing systems.

## 2.4 A note on desiderata 3 - plug-in readiness and local explanations

One of many challenges of building a post-processing interpretion method is to ensure that the explanation is truthful to the model's behavior. By "truthful" we mean that explanations are roughly consistent with model's internal state. For example, if the explanations are created completely independently of the model (11), it has high probability of having such inconsistencies.

The plug-in readiness desiderata poses interesting challenge for explanations to remain consistent with the model's behavior. Recently, there has been work showing that saliency methods contains such inconsistencies. For instance, (14) show that saliency methods are vulnerable to constant

shift in input that does not affect classification. It has also shown that the methods can be easily tricked (9).

One way to improve consistency between explanations and the model's reasoning is to use the generated explanation as an input, and check the network's output for validation. This is typically used in perturbation-based interpretability methods (16; 20). These methods perturb the input, and use the network's response to generate explanations. They maintain the consistency either locally or globally[1] by construction. TCAV is a type of perturbation method.

Even a truthful explanation may be misleading if it is only locally truthful (20). For example, since the importance of features only needs to be truthful in the vicinity of the data point of interest, there is no guarantee that the method will not generate two completely conflicting explanations. These inconsistencies may result in decreased user trust at a minimum. On the other hand, making a globally truthful explanation may be difficult as the networks decision boundaries may be complex. TCAV produces globally explanations, and uses model's output to generate explanations to maintain consistency between explanations and the model's reasoning.

## 3 TCAV Method

We introduce a method that allows for global linear interpretability of a highly flexible class of models, namely deep feedforward neural networks trained on classification tasks. As a form of explanation, TCAV uses concepts that are provided by users, instead of using predetermined set of input features or input classes. These concepts are tied to real-world data that represents concepts of interest. Users express their concepts of interest using examples — a set of data points exemplifying the concept. These concepts enable testing the relative importance of concepts used in classification.

Informally, the key idea is the following: although feedforward networks learn highly nonlinear functions there is evidence that they work by gradually disentangling concepts of interest, layer by layer (2; 3). It has also been shown that representations are not necessarily contained in individual neurons but more generally in linear combinations of neurons (19; 27). Thus the space of neuron activations in layers of a neural network may have a meaningful global linear structure. Furthermore, if such a structure exists, we can uncover it by training a linear classifier mapping the representation in a single layer to a human selected set of concepts.

We now formalize this intuition. First, we formally define a *concept activation vector*. Let us imagine that an analyst is interested in a given concept $C$ (e.g., striped textures) and has gathered two sets of data points, $P_C$ and $N$, that represent positive and negative examples of this concept (say, photos of striped objects, versus a set of random photos). There is a lot of flexibility in how to choose $N$, we often choose a random set of images of the same size as $P_C$. Here we represent the input to the network as vector in $\mathbb{R}^n$, and $P_C, N \subset \mathbb{R}^n$. Consider a layer $l$ of the feedforward network consisting of $m$ neurons. Then running inference on an input example and looking at the activations at layer $l$ yields a function $f_l : \mathbb{R}^n \to \mathbb{R}^m$. For a set of inputs $X \subseteq \mathbb{R}^n$ we denote by $f_l(X)$ to be the set of layer activations $\{f_l(x) : x \in X\}$. Note that for convolutional networks we view a layer in it's flattened form, thus a layer of width $w$, height $h$, and $c$ channels becomes a flat vector of $m = w \times h \times c$ activations.

The two sets $P_C$ and $N$ then naturally give rise to two sets of activation vectors in layer $l$, namely $f_l(P_C)$ and $f_l(N)$. We can then train a linear classifier on the binary classification task of distinguishing between these two sets. The weights of this classifier are an element $v_C^l \in \mathbb{R}^m$. We call this the concept activation vector for the concept $C$.

A variation on this idea is a *relative* concept activation vector. Here the analyst selects two sets of inputs that represent two different concepts, $C$ and $D$. Training a classifier on $f_l(P_C)$ and $f_l(P_D)$ yields a vector $v_{C,D}^l \in \mathbb{R}^m$. The vector $v_{C,D}^l$ intuitively defines a $1 - d$ subspace in layer $l$ where the projection of an embedding $f_l(x)$ along this subspace measures whether $x$ is more relevant to concept $C$ or $D$.

---

[1]The global explanation means explanations that are globally true, while local explanations mean explanations that are only locally (i.e., a data point and its neighbors) true, following definitions from (6).

A key benefit of this technique is the flexibility allowed in choosing the set $P_C$. Rather than being tied to the class labels in the training data, an analyst can–after training–select sets that correspond to any concept of interest.

We now describe two ways that this technique be used to interpret a feedforward neural network trained on an image classification task. First, we show relative importance testing between $M$ number of concepts, $P_{Ci}$, where $i \in M$. In this case, each $v_C^l$ is learned by treating all $P_{Ci}, i \neq j$ as $N$. Second, we can also test one concept, $P_C$, against a random concept, where $N$ are set of random pictures. In the next section we discuss the results of experiments with these methods.

## 3.1 APPLICATION: TESTING THE IMPORTANCE OF CONCEPTS IN CLASSIFICATION

The real value of CAVs comes from the fact that they may be used to test the relative importance of concepts. With CAVs, we can formulate generating explanation as a task of performing two-tailed statistical significance test (in particular, z-test). Given samples of class images (e.g., zebra pictures) and two concept vectors A and B, we perform two-tailed z-testing to invalidate the null hypothesis that there is no difference in importance of concepts A and B for the class. We perform this testing for each pair of concepts. For example, an analyst might ask about a photo of a zebra, "was the presence of stripes in the image relevant to the model's classification of the image as a zebra?" In some cases, with tedious effort in a photo retouching program, it might be possible answer questions like this directly. However, CAVs provide a faster and more general technique.

Consider an input $x$ and concept $C$ of interest. At inference time, this will give rise to activations $f_l(x)$ at layer $l$. We can then modify this vector of activations in various ways using the concept vector $v_C^l$. For example, we might create a new vector $w^- = f_l(x) - v_C^l$, which might be interpreted as a counterfactual of "$x$ with less of concept $C$". Performing inference starting at layer $l+1$ based on $w^-$ will give a new classification result $y_w$. We can also create a new vector $w^+ = f_l(x) + v_C^l$, which might be interpreted as a counterfactual of "$x$ with more of concept $C$".

Thus to test the relevance of stripes to the classification of a zebra, we would either add or subtract a 'stripe' concept activation vector from the zebra image embedding, run forward propagation the resulting embedding and examine the output of the network. Large changes in the probability of zebra would indicate that stripes played a key role in the classification. Simply adding the vector is a bit ad-hoc, especially when the norm of the vector may depend on how exactly the CAV was trained. However, we found this naive method to empirically yield results that were consistently semantically meaningful. We also note that this addition is loosely related to directional derivative – saliency maps take the derivative of the logits with respect to each pixel, while our work takes derivatives with respect to a concept direction. Future work should explore more principled methods, to measure relevance of concepts to classification.

Quantitatively, the metric of the influence of the concept $C$ to class $k$ can be measured by

$$I_w^{up} = \frac{1}{N} \sum_i^N \mathbb{1}\left(p_k(y) < p_k(y_w)\right) \qquad I_w^{down} = \frac{1}{N} \sum_i^N \mathbb{1}\left(p_k(y) > p_k(y_w)\right) \qquad (1)$$

where $p_k(y)$ represents the probability of class $k$ of prediction $y$ and $i = \{1, 2, \ldots, N\}$ where $N$ is the number of images of the class for inspection (e.g., zebra). Intuitively speaking, $I_w^{up/down}$ measures the ratio of data points that become 'more/less like class $k$' after the modification with concept vector $v_C^l$.

**We can perform statistical significance testing in order to quantify the concept importance; we can test a hypothesis that one concept is more/less important than another concept. In other words, the null hypothesis that no color is significant. To test this hypothesis we can do z-testing on the measured importance values, and ask the question what is the probability that random concept vectors would observe the measured difference.**

## 3.2 APPLICATION: WHERE ARE CONCEPTS LEARNED?

As a simple demonstration of the value of CAVs, we describe a technique for localizing where a concept is disentangled in a network. A common view of feedforward networks is that different concepts are represented at different layers. For example, many image classification networks are

said to detect textures at low levels, with increasingly complex concepts recognized at later layers. Typically evidence for this view comes from various methods of reverse-engineering (3; 30; 17) the behavior of individual neurons in the network.

Using concept activation vectors, however, we can approach this question from the opposite direction. We can pick out a particular concept $C$ in advance, and consider the concept vectors $v_C^1, v_C^2, \ldots, v_C^k$ for each layer in the network. The accuracies of the linear classifiers, $a_C^1, a_C^2, \ldots, a_C^k$ then provide a measure of how well disentangled the concept is in each layer. This may be viewed as a generalization of the ideas in (2; 3; 19). In the next section we describe the results of an experiment based on this idea.

## 4 RESULTS

In this section, we first show evidence that the learned CAV indeed detect the concepts of interest. Then we show the results for hypothesis testing using these CAVs, and the insights one can gain from it.

All our experiments are based on the model from (26) using the publicly available model parameters from (1). The pictures used to learn the CAVs are collected from the latest ImageNet Fall 2011 release (21). Each concept vector is learned using between 30-500 pictures as input. We intentionally chose not to use all the available pictures in order to mirror realistic scenarios where users may not be able to collect a large amount of data. The class 'arms' only had 30 pictures (all manually collected), since ImageNet does not include this as a part of the dataset. The pictures used to learn texture concepts are taken from the (3) data set. In order to learn concept vectors for colors, we generated 500 pictures synthetically by generating the color channels randomly. We provide experiments both where CAV's are trained to distinguish between a set of concepts (e.g. red, yellow, green, blue) and when one CAV is trained per concept (with $N$ chosen to be a random collection of images).

### 4.1 EVIDENCE OF THE VALIDITY OF LINEARITY ASSUMPTION

In this section we describe experiments that indicate our linearly learned CAVsalign with their intended concepts. First, the linear maps used to construct the CAVsare accurate in predicting the concepts. The point in the network where these concepts in the networks are learned (i.e., when accuracy becomes high) is consistent with general knowledge about the representations learned by neural networks. Low level features such as edges and textures are detected in the early layers and higher level concepts are only reliably detected in the upper layers. Next, we use activation maximization techniques (18) to visualize each of the CAVs and observe that the patterns are consistent with the provided concept. Finally, we show the top $k$ images that are most similar in terms of cosine similarity to each CAV for further qualitative confirmation.

#### 4.1.1 WHERE EACH CONCEPT IS LEARNED.

Figure 1 shows the accuracy of the linear classifiers at each layer in the network for each type of CAV. Each classifier is trained to distinguish one concept from other concepts (e.g., textures1 set contains 'stripes', 'zigzagged' and 'dotted' texture concepts). Overall, we observe high accuracy as measured by a held out test set of size 1/3 that of the training size. This is evidence that the given concepts are linearly separable in many layers of the network.

Note that the accuracy of more abstract CAV (e.g., objects) increases in higher layers of the network. The accuracy of a simpler CAV, such as color, is high throughout the entire network. This agrees with prior work on visualizing the learned representations at different layers of neural networks (30).

This experiment does not yet show that these CAVs align with the concepts that makes sense semantically to humans. We demonstrate this with the next set of experiments.

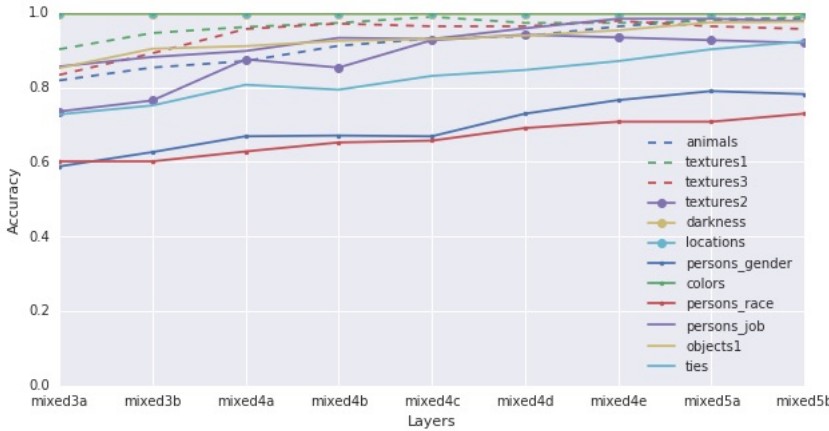

Figure 1: Where each of the CAVs are learned. Accuracy at each layer for CAV. Simple concepts (e.g., colors) achieve high performance in all layers than more abstract concepts (e.g. persons, objects)

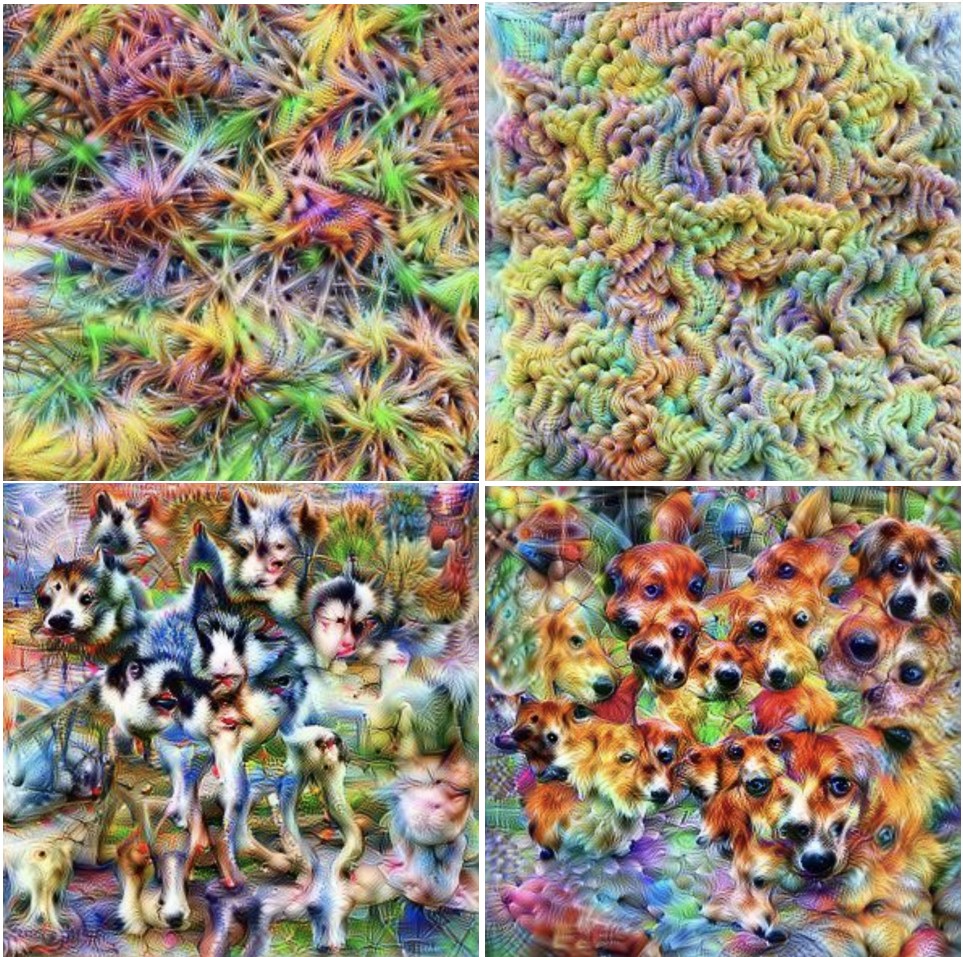

Figure 2: Deepdreamed fibrous texture, knitted texture, corgis and Siberian huskey CAVs.

### 4.1.2 EMPIRICAL DEEPDREAM: ACTIVATION MAXIMIZATION IN THE DIRECTION OF THE CAVS

In this section, we use the activation maximization technique (17) to visualize the learned representations in the direction of the CAV. We use the same techniques from (18). As typically done, we use a random image as a starting point (17) for the optimization to avoid choosing an arbitrary image as a starting point. Figure 2 shows highly recognizable features, such as knitted textures, and corgi faces.

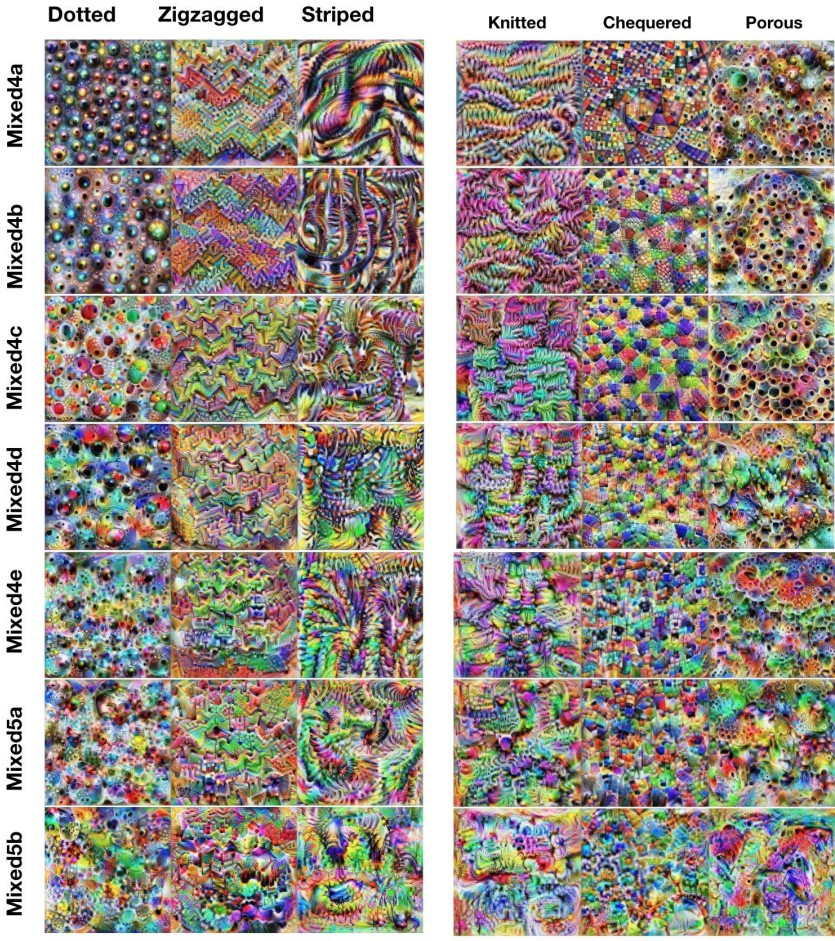

Figure 3: Deepdreamed CAV texture concepts for each layer (each row) in inception. We can identify textures better in the mid-layer (mixed4d), then later layers.

We can also observe how each concept is represented as the depth increases. Images in Fig. 3 show the results for sets of textures. Interestingly, there is a layer (around mixed 4d) where textures are clearly identifiable, after which they become less and less recognizable.

The left images in Fig. 4 show the results for set of colors — green, blue, yellow and red (each column) and for the set of layers (lower to higher layers from top to bottom). The training set for the color CAVs are generated by simply replacing each RGB channel with randomly sampled values around 0.5, while leaving other channels empty. Note how the color CAVs also contain many random textures, this is perhaps not surprising - there is no reason to expect any direction in a layer to be associated purely with one concept while being orthogonal to all other concepts.

The right image in Fig. 4 shows higher level concepts. The columns represent zebra, siberian huskies, and corgis. It is interesting to note how the zebra CAVs include textures suggestive of water, trees, and grass. This suggests that the model associates all of these concepts with the classi-

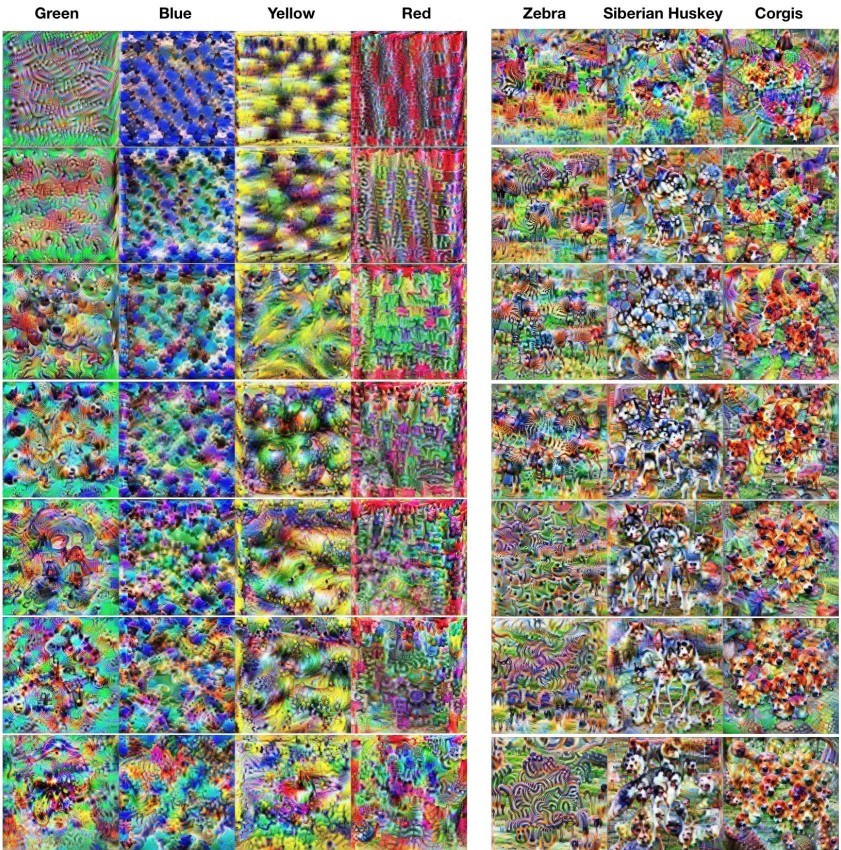

Figure 4: Deepdreamed CAV color and animal concepts for each layer (each row) in inception. Zebra CAVs include textures suggestive of water, trees and grass.

fication of zebra and is likely a learned bias that results from the background of zebra pictures in the training data.

These visualizations provide some qualitative confirmation that the learned directions align with the given concepts. This also shows that DeepDream may be combined with TCAV in order to identify and visualize interesting directions in a layer. Prior work on DeepDream has typically chosen neurons or linear combinations of neurons at random to visualize. The next section provides further evidence that the CAVs are indeed aligned with the concept of interest using real data.

### 4.1.3 QUALITATIVE CONFIRMATION: PICTURES THAT ARE SIMILAR TO CAV

In order to qualitatively confirm that the learned CAV aligns with meaningful concepts, we compute cosine similarity between a set of pictures (all from one class) to the CAV. Fig. 5 shows that the top corgi images that aligns with striped CAV selects pictures with striped objects (e.g., a striped tube or vertical blinds in the background). Thus being similar to the striped CAV meant being highly similar to one of the other concepts in the CAV training set. We also show that CAVs for more abstract concepts can be constructed (e.g., CEO). Recall that many of these learned concepts are not classes that the NN learned to classify.

We also observe that if there is not much relation between pictures and the concept, the cosine similarity is low. This experiment qualitatively supports our argument that the alignment between CAVs and the meaningful concepts. Note that the images least similar to striped appear with a checkered or knitted pattern, likely due to the fact that the striped CAV was trained relative to checkered and knitted images.

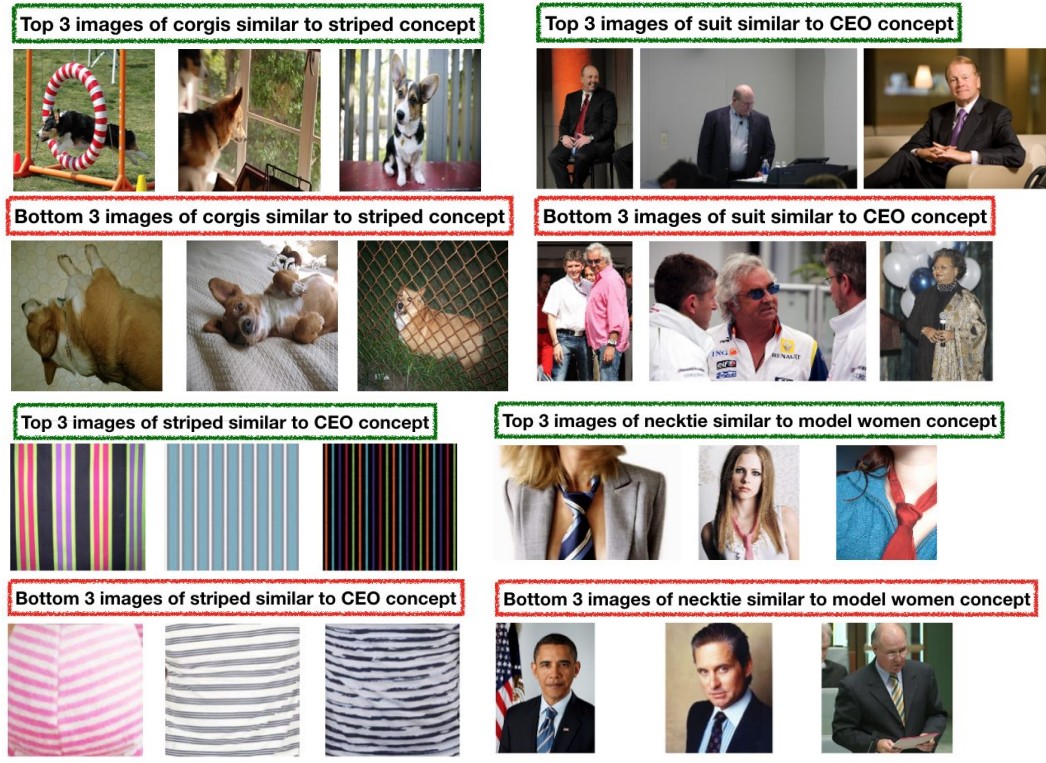

Figure 5: Top and bottom 3 similar pictures of each concept

When using TCAV, in order to ensure that the provided examples were sufficient to learn CAV, we recommend users to perform this qualitative test by checking cosine similarity with each concepts to the class.

## 4.2 TESTING RELATIVE IMPORTANCE OF CONCEPTS

In this section, we describe how concept activation vectors may be used for quantitative testing of the relative importance of concepts.

In Fig 6, we show $I_{w+}^{up}$ for eight classes. We do not list $I_{w-}^{up}$, as they always appear to be the inverse of $I_{w+}^{up}$. This means that when CAVs are added, the probability of the class rarely stays the same. This confirms that adding or subtracting $v_C^l$ clearly impacts the classification task. In particular the $v_C^l$ direction is aligned with the direction that measures the probability of the target class.

Note that adding random directions should be expected to change the probability of the target class in some way. We find, however, that for several CAV's which are semantically relevant to the class (e.g., red to fire engine) the CAVdirection will consistently increase the probability of class for most images of this class. On the other hand, we show that random directions (e.g., a concept learned from random set of images) tend to be much less consistent. This difference allows us to perform a more rigorous hypothesis test in a following section.

This testing identifies relationships between concepts and the target class that agree with human intuition (e.g., fire engines are red, cucumbers are bumpy, zebras are striped, CEOs wear suits). In the next section, we describe insights we gained from these tests.

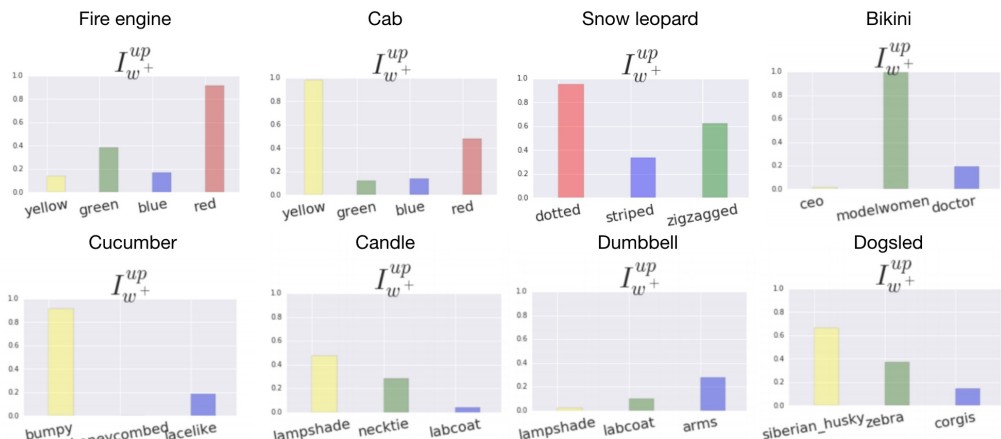

Figure 6: Testing Relative Importance of Concepts

## 4.3 Gaining insights

In this section, we show results that confirm common-sense intuitions about training images, as a kind of sanity check that TCAV provides valid results. We then describe results that surprised us, but lead to insights about the dataset that we did not initially have.

In Fig 6, the yellow color was more influential to cab class than any other colors. For people who have lived in cities where this is the standard color for taxis, this is no shock — and indeed, most ImageNet pictures of cabs are yellow. Similarly, the 'women' concept is important to 'bikini' class, likely due to the fact that training images have humans wearing the bikini rather than pictures of the clothing. More subtly, we also discovered that the 'model women' concept is important to 'bikini' This lead us to realize that the most of 'bikini' class pictures feature professional models, typically very slim, posing in their bikinis — probably not well representative samples of bikinis in the real world. Note that the network was trained with bikini and cab classes, and the concept 'yellow', 'women' and 'model women' are not part of classes used to train this network[2].

The graph for 'dumbbell' class in Fig 6 shows that 'arms' concept was more important to predict dumbbell class than other concepts. This finding is consistent with previous qualitative findings from (17), where they discovered that the DeepDream picture of a dumbbell also showed an arm holding it. TCAV allows for quantitative confirmation of this previously qualitative finding. Moreover, unlike all other concepts in this figure, we only collected 30 pictures of each concept (ImageNet did not have arms as a label). Despite the small number of examples, the TCAV method is able to identify this relationship.

The flexibility of the TCAV method makes it easy to explore a network for other surprising associations. For example, we saw that 'baby' concept was important to 'school bus' class. It turns out that some school bus pictures include young children riding or standing in front of the bus. Another discovery was to find that 'red' concept is important for cucumber class. We believe this due to the fact that a number of cucumber pictures include other red vegetables, such as tomato and carrot.

We believe that TCAV can be useful in discovering these insights for many applications, including for example, to improve fairness in the model.

## 4.4 Random image comparisons to filter spurious results

When constructing the CAV, the choice of negative samples may cause spurious suggestions that a concept is relevant to a class when it actually is not. In other words, the learned CAV may accidentally be aligned with something related to the class, and cause high $I_{w^+}^{up}$. For instance, when we

---

[2]Only a subset of labels from ImageNet are used in training this network. 'women' concept was not part of prediction class, and we collected these pictures from ImageNet dataset (21)

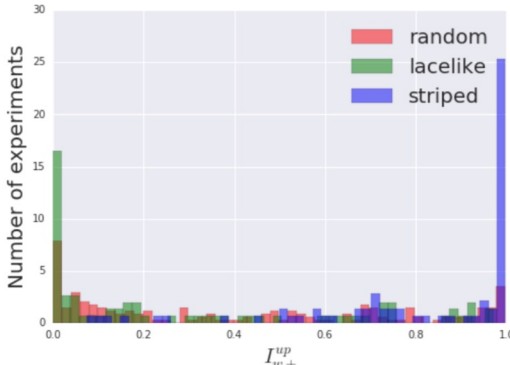

Figure 7: Histogram of the mean of $I_{w^+}^{up}$ for the relationship of the zebra class with striped, lace-like and random concepts.

tested unrelated concepts and classes, such as zebra to a set of textures, honeycombed, lace-like and bumpy textures, we found that lace-like concept shows high $I_{w^+}^{up}$.

One might argue that lace is vaguely related to a zebra's stripes, but in fact even if we choose both $P_C$ and $N$ to be independent random sets of images we still often observe $I_{w^+}^{up}$ to be relatively high. This is not surprising: there are directions in a layer $l$ which are highly aligned with the zebra concept, and randomly chosen directions will typically have small but non-zero projection along these directions.

One way to filter out these spurious results is to do statistical testing against random concepts. Using different sets of random images for the negative class and striped images for $P_C$, we make 50-100 CAVs, all of which represent concept 'striped'. For each 'striped' CAV we can measure $I_{w^+}^{up}$. We can also generate a set of 'random' CAVs by choosing random images for both $P_C$ and $N$. Then we performed a $z$-test to see if the mean $I_{w^+}^{up}$s from striped CAVs are statistically different from the mean $I_{w^+}^{up}$ of random CAVs. We can successfully filter out some spurious correlations, including lace-like concepts with zebras by using this method, see the histogram 4.4.

## 5 CONCLUSION

We have introduced the notion of a "concept activation vector," or CAV, which is a flexible way to probe the internal representation of a concept in a classification network. Since CAVs may be defined via a set of example inputs, rather than custom coding, they are well suited to use by non-experts. We then described a technique (Testing with CAVs, or TCAV) for quantifying the relation between a CAV and a particular class. The TCAV technique allows us to provide quantitative answers to questions such as, "How important are the stripes to the classification of a zebra?"

To provide evidence for the value of the TCAV technique, we described a series of experiments which supported common-sense intuition, for example, that stripes are indeed important to the identification of zebras. In addition, we used the DeepDream technique to create images whose internal representations approximate certain CAVs. The resulting pictures were strongly evocative of the original concepts. Finally, we described how the TCAV technique may be used to find associations between concepts, both obvious ("yellow" and "taxi") and non-obvious ("red" and "cucumber").

In addition to analyzing a single network, TCAV can be also used to compare and contrast a pair of networks. For example, one can compare the relative importance of concepts to determine how the different choices of training process or architecture influences learning of each concept. Based on the results, users can perform model selection based on the concepts that are more or less important for the task.

An interesting direction for future work may be to explore applications of using CAVs to adjust the results of a network during inference time. Adding a scalar multiple of a CAV to the activations of an intermediate layer can, as shown in our experiments, allow us to deemphasize or enhance conceptual

aspects of an input. One potential application, for example, might be to reduce bias the network has learned from training data.

ACKNOWLEDGMENTS

We would like to thank Chris Olah, Alexander Mordvintsev and Ludwig Schubert for generously allowing us to use their code for DeepDream.

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

top 3 images of corgis similar to knitted concept

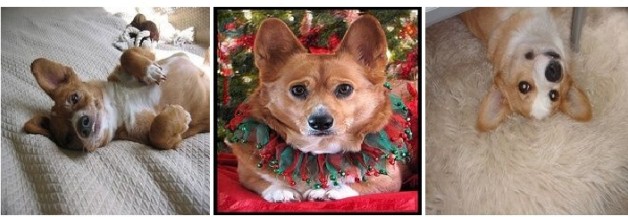

bottom 3 images of corgis similar to knitted concept

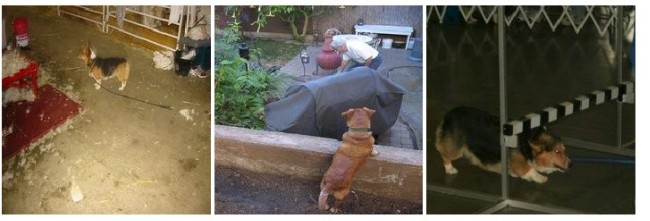

top 3 images of salmon similar to knitted concept

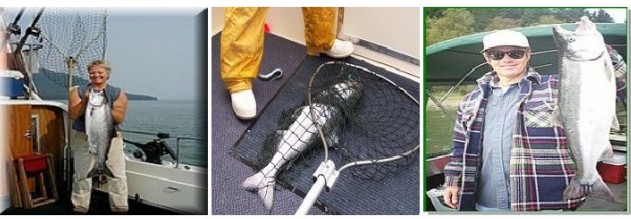

bottom 3 images of salmon similar to knitted concept

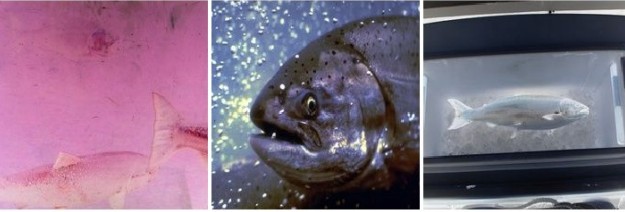

top 3 images of corgis similar to dotted concept

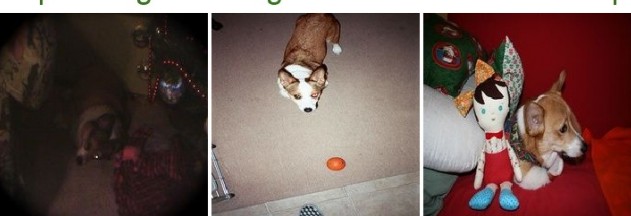

bottom 3 images of corgis similar to dotted concept

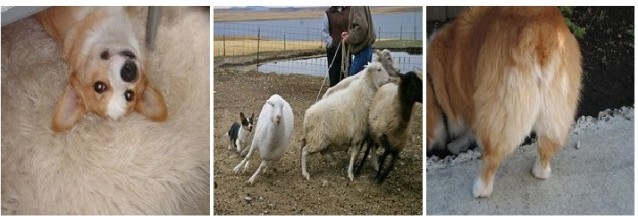

# 6 APPENDIX

top 3 images of salmon similar to dotted concept

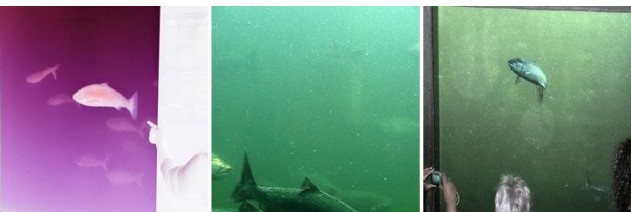

bottom 3 images of salmon similar to dotted concept

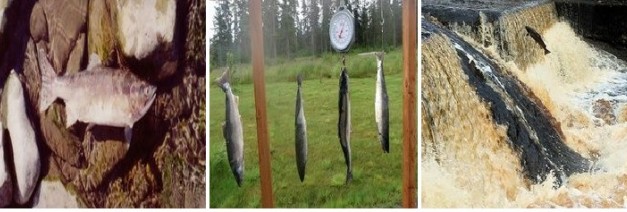

top 3 images of zebra similar to striped concept

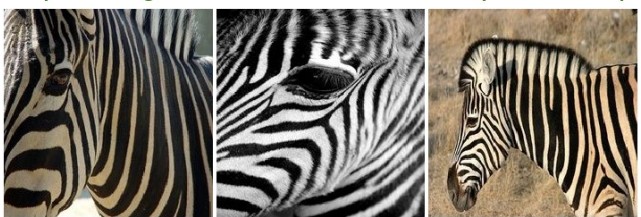

bottom 3 images of zebra similar to striped concept

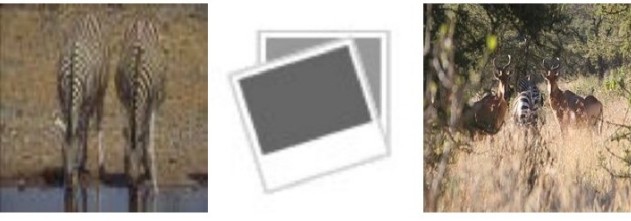

top 3 images of salmon similar to striped concept

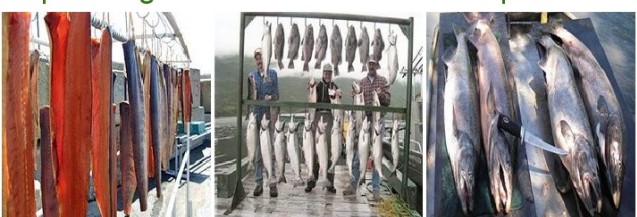

bottom 3 images of salmon similar to striped concept

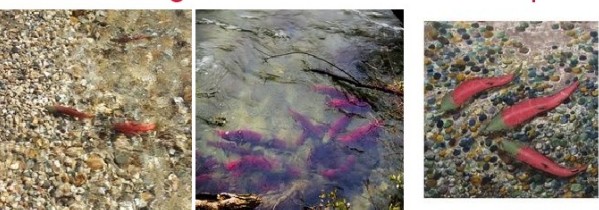

**top 3 images of corgis similar to porous concept**

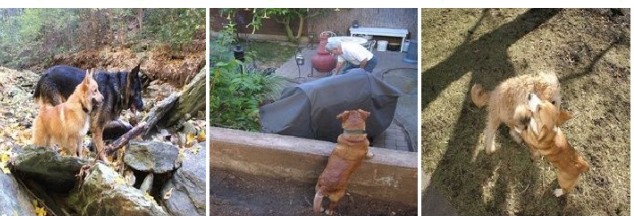

**bottom 3 images of corgis similar to porous concept**

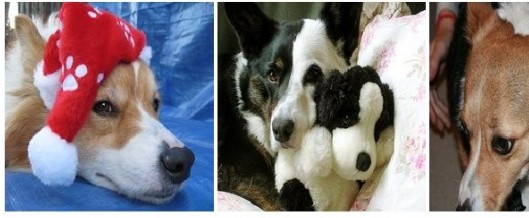

**top 3 images of salmon similar to porous concept**

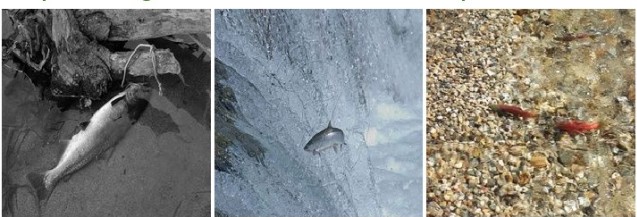

**bottom 3 images of salmon similar to porous concept**

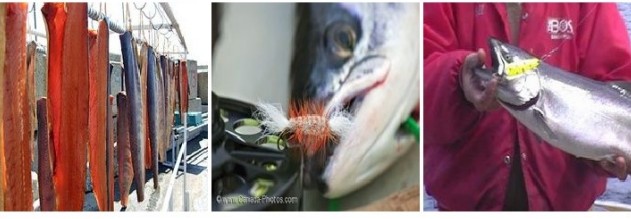

**top 3 images of CEO similar to labcoat concept**

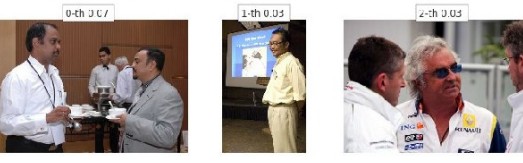

**bottom 3 images of CEO similar to labcoat concept**

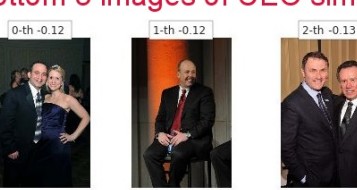

### top 3 images of doctor similar to labcoat concept

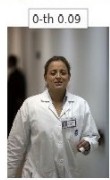 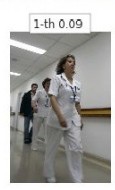 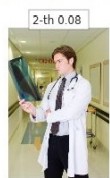

### bottom 3 images of doctor similar to labcoat concept

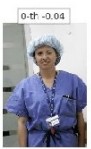 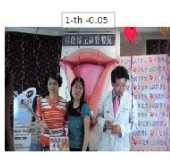 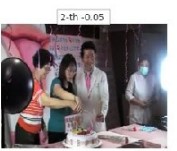

### top 3 images of boss similar to labcoat concept

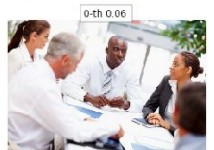 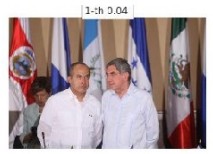 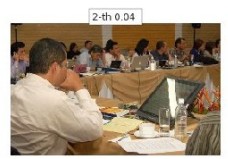

### bottom 3 images of boss similar to labcoat concept

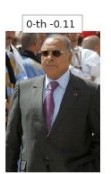 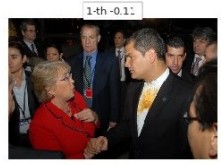 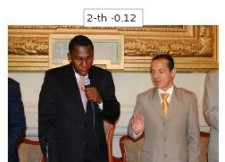

### top 3 images of model women similar to labcoat concept

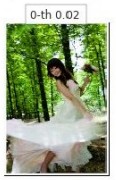 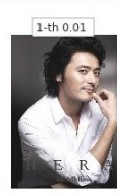 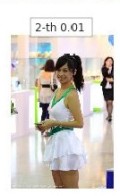

### bottom 3 images of model women similar to labcoat concept

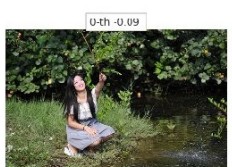 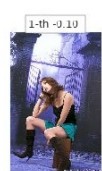 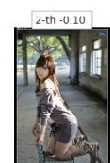

top 3 images of CEO similar to suit concept

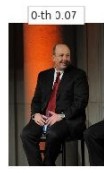 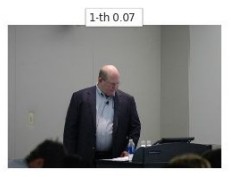 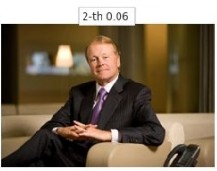

bottom 3 images of CEO similar to suit concept

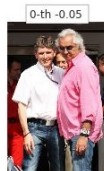 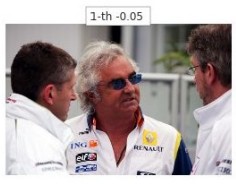 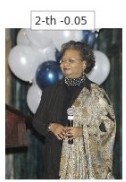

top 3 images of head nurse similar to suit concept

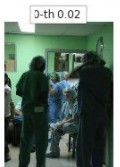 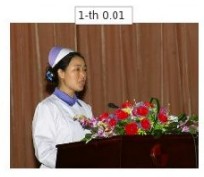 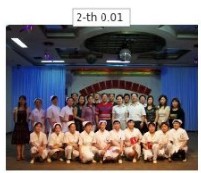

bottom 3 images of head nurse similar to suit concept

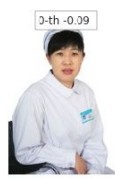 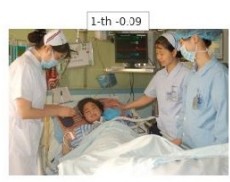 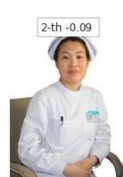

top 3 images of boss similar to suit concept

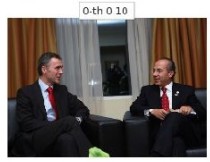 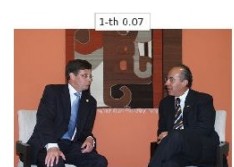 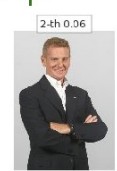

bottom 3 images of boss similar to suit concept

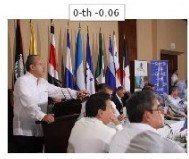 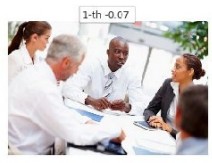 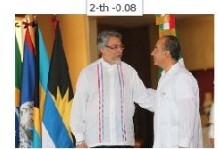

top 3 images of model women similar to suit concept

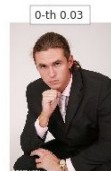 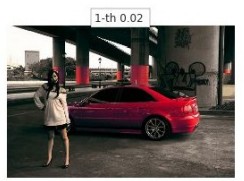 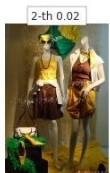

bottom 3 images of model women similar to suit concept

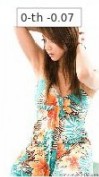 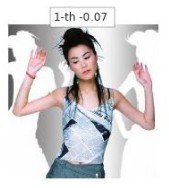 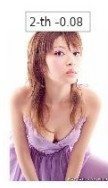

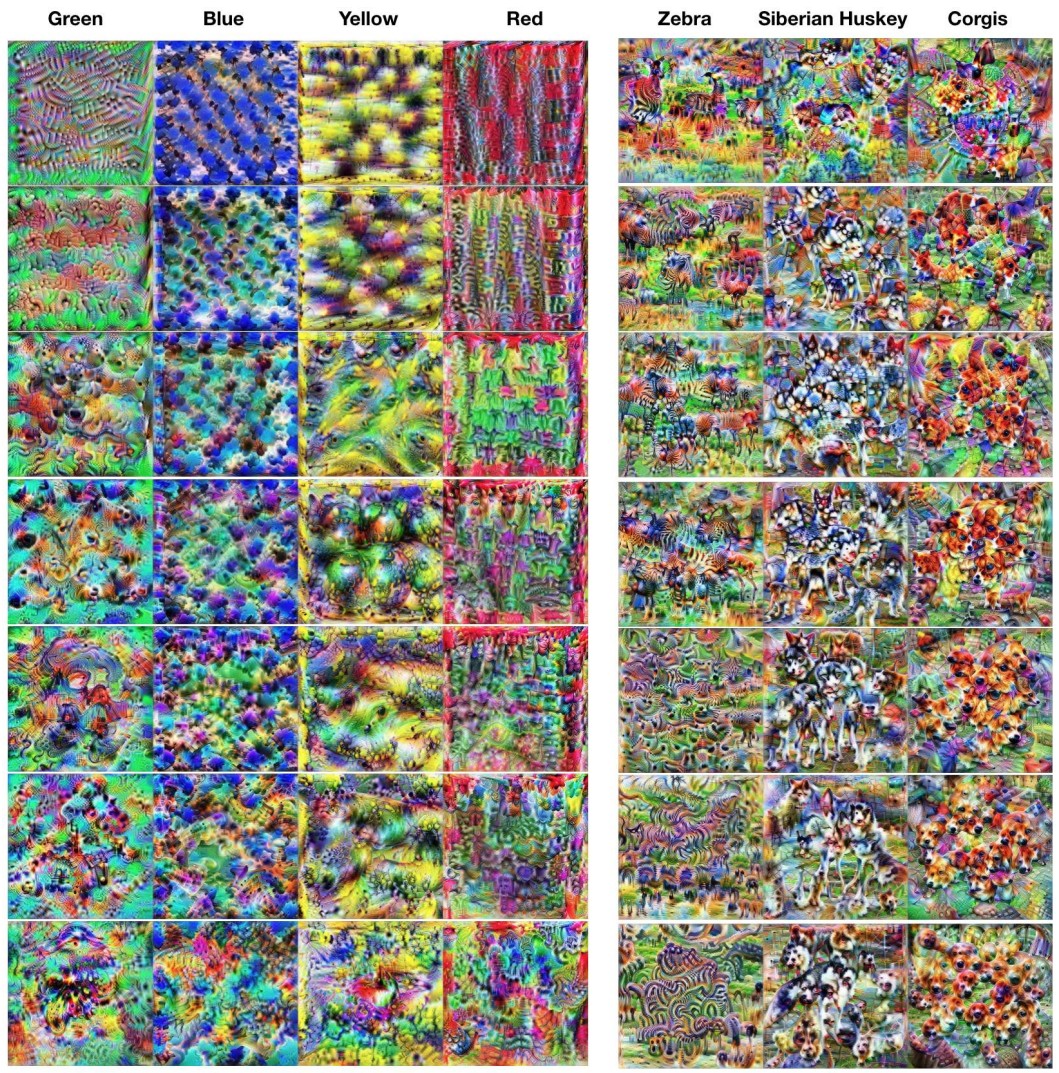

