# OpenReview forum: "TCAV: Relative concept importance testing with Linear Concept Activation Vectors"
_ICLR.cc/2018/Conference — Reject_

### Official Review · AnonReviewer1 · 2017-11-27
**Concept activation vectors make sense for interpretability, but the presentation and evaluation need improvement.**

**Rating:** 4
**Confidence:** 4

**Review:**

Summary
---
This paper proposes the use of Concept Activation Vectors (CAVs) for interpreting deep models. It shows how concept activation vectors can be used to provide explanations where the user provides a concept (e.g., red) as a set of training examples and then the method provides explanations like "If there were more red in this image then the model would be more likely to classify it as a fire truck."

Four criteria are enumerated for evaluating interpretability methods:
1. accessibility: ML background should not be required to interpret a model
2. customization: Explanations should be generated w.r.t. user-chosen concepts
3. plug-in readiness: Should be no need to re-train/modify the model under study
4. quantification: Explanations should be quantitative and testable

A Concept Activation Vector is simply the weight vector of a linear classifier trained on some examples (100-500) of a user-provided concept of interest using features extracted from an intermediate network layer. These vectors can be trained in two ways:
1. 1-vs-all: The user provides positive examples of a concept and all other existing training data is treated as negatives
2. 1-vs-1: The user provides sets of positive and negative examples, allowing the negative examples to be targeted to one category

Once a CAV is obtained it is used in two ways:
First, it provides further verification that higher level concepts tend to be "disentangled" in deeper network layers while low level concepts are "disentangled" earlier in the network. This work shows that linear classifier accuracy increases significantly using deeper features for higher level concepts but it only increases marginally (or even decreases) when modeling lower level concepts.

Second, and this is the main point of the paper, relative importance of concepts w.r.t. a particular task can be evaluated. Suppose an image (e.g., of a zebra) produces a feature vector f_l at layer l and v_l is a concept vector learned to classify the presence of stripes from layer l features. Then the probability the model assigns to the zebra class can be evaluated using features f_l and then f_l + v^c_l. If the latter probability is greater then adding stripes will increase the model's confidence in the zebra class. Furthermore, the method goes on to measure how often stripes increase zebra confidence across all images. Rather than explaining the network's decision for a particular image, this average metric measures the global importance of the stripes concept for zebra. The paper reports examples of the relative importance of certain concepts with respect to others in figure 5.


Pros
---

The paper proposes a simple and novel idea which could have a major impact on how deep networks are explained. At a high level the novelty comes from replacing the gradient (or something similar) used in saliency methods with a directional derivative. Users can align the direction to any concept they find relevant, so the concept space used to explain a prediction is no longer fixed a-priori (e.g. to pixels in the input space). It can adapt to user suspicions and expectations.


Cons
---

Concerns about story/presentation:

* The second use of CAVs, to test relative importance of concepts, is basically an improved saliency method. It's advantages over other saliency methods are stated clearly in 2.1, but it should not be portrayed as fundamentally different.

The two quantities in eq. 1 can be thought of in terms of directional derivatives. To compute I_w^up start by computing a finite differences approximation of directional derivative of the linear classifier probability p_k(y) with respect to layer l features in the direction of the CAV v_C^l. Call this quantity g_i (for the ith example). Then I_w^up is the average of 1(g_i > 0) over all examples. I think the notion of relative importance used here is basically the idea of a directional derivative.

This doesn't change the contribution of the paper but it should be mentioned and section 2.1 should be changed so it doesn't suggest this method is fundamentally different than saliency methods in terms of criteria 4.

* Evaluation and Desiderata 4: The fourth criteria for interpretability laid out by the paper says an explanation should be quantitative and testable. I'm not sure exactly what this is supposed to mean. I see two ways to interpret the quantitative criterion.

One way to interpret the "quantifiability" criterion is to say that it requires explanations to be presented as numeric values. But most methods do this.  In particular, saliency methods report results in terms of pixel brightness (that is a numeric quantity) even though humans may not know how to interpret that correctly. I do not think this is what was intended, so my second option is to say that the criterion requires an explanation be judged good or bad according to some quantitative metric. But this paper provides no such metric. The explanations in figure 5 are not presented as good or bad according to any metric.

While it is significant that the method meets the first 3 criteria, these do not establish the fidelity of the method. Do humans generalize these explanations to valid inferences about model behavior? Maybe consider some evaluation options from section 3 of Doshi-Velez and Kim 2017 (cited in the paper).

* Section 4.1.1: "This experiment does not yet show that these concept activation vectors align with the concepts that makes sense semantically to humans."

Isn't test set accuracy a better measure of alignment with the human concept than the visualizations? Given a choice between a concept vector which produced good test accuracy and poor visualizations and another concept vector which produced poor test accuracy and good visualizations I would think the one with good test accuracy is better aligned to the human concept. I would still prefer a concept vector which satisfies both.

* Contrary to the description in section 2.2, I think DeepDream optimizes a natural image (non-random initialization) rather than starting from a random image. It looks like these visualization start from a random initialization. Which method is used? Maybe cite this paper, which gives a nice overview: "Multifaceted Feature Visualization: Uncovering the Different Types of Features Learned By Each Neuron in Deep Neural Networks" by Nguyen et. al. in the Visualization for Deep Learning workshop at ICML 2016

* In section 4.1.3 I'm not quite sure what the point is. Please state it more clearly. Is the context class the same as the negative set used to train the classifier? Why should it be different/harder to sort corgi examples according to a concept vector as opposed to sorting all examples according to a concept vector? This seems like a useful way of testing to be sure CAV's represent human concepts, but I'm not sure what context concepts like striped/CEO provide.

* Relative vs absolute importance and user choice: Section 4.2 claims that figure 5 shows that a CAV "captures an important aspect of the prediction." I would be a bit more careful about the distinction between relative and absolute here. If red makes images more probably fire trucks then it doesn't necessarily mean that red is important for the fire truck concept in an absolute sense. Can we be sure that there aren't other concepts which more dramatically affect outputs? What if a user makes a mistake and only requests explanations with respect to concepts that are irrelevant to the class being explained? Do we need to instruct users on how to best interpret the explanation?

* How practical is this method? Is it a significant burden for users to provide 100-500 images per concept? Are the top 100 or so images from a search engine good enough to specify a CAV?


Minor missing experimental settings and details:

* Section 3 talks about a CAV defined with respect to a non-generic set D of negative examples. Is this setting ever used in the experiments or is the negative set always the same? How does specifying a narrow set of negatives change the CAV for concept C?

* I assume the linear classifier is a logistic regressor, but this is never stated.

* TCAV measures importance/influence as an average over a dataset. This is a validation set, right? For how many of these images are both the user concept and target concept unrelated to the image content (e.g., stripes and zebra for an image of a truck)? When that happens is it reasonable to expect meaningful explanations? They may not be meaningful because the data distribution used to train the CAV probably does not even sparsely cover all concepts in the network's train set. (related to "reference points" in "The (Un)reliability of Saliency Methods" submitted to ICLR18)

* For relative importance testing it would be nice to see a note about the step size selection (1.0) and experiments that show the effect of different step sizes. Hopefully influence is monotonic in step size so that different step sizes do not significantly change the results.

* How large is the typical difference between p_k(y) and p_k(y_w) in eq. 1? If this difference is small then is it meaningful? Are small differences signal or noise?


Final Evaluation
---
I would like to see this idea published, but not in its current form. The method meets a relevant set of criteria that no other method seems to meet, but arguments set forth in the story need some revision and the empirical evaluation needs improvement, especially with respect to model fidelity. I would be happy to change my rating if the above points are addressed.

---

### Official Review · AnonReviewer2 · 2017-12-12
**Interesting work, but can be improved significantly in terms of clarity, claims, and evaluation**

**Rating:** 4
**Confidence:** 3

**Review:**

Strengths:
1. This paper proposes a novel method called Concept Activation Vectors (CAV) which facilitates interpretability of neural networks by explaining how much a specific concept influences model predictions.
2. The proposed method tries to incorporate multiple desiderata, namely, accessibility to non ML experts, customizability w.r.t. being able to explain any concept of interest, plug-in readiness i.e., providing explanations
without requiring retraining of the model.

Weaknesses:
1. While this work is conceptually interesting, the technical novelty and contributions seem fairly minimal.
2. The presentation of this paper is one of its weakest points. The organization of the content is quite incoherent. The paper also makes a lot of claims (e.g., hypothesis testing) which are not really justified.
3. The experimental evaluation of this paper is quite rudimentary. Lots of details are missing.

Summary: This paper proposes a novel framework for explaining the functionality of neural networks by using a simple idea. The intuition behind the proposed approach is as follows: by using the weight vectors of linear classifiers, which take as inputs the activation layer outputs of a given neural network (NN) model and predict the concepts of interest, we can understand the influence of specific concepts of interest on the NN model behavior. The authors claim that this simple approach can be quite useful in providing explanations that can be useful for a variety of purposes including testing specific hypothesis which is never really demonstrated or explained well in the paper. Furthermore, lot of details are lacking in both the experimentation section and the methods section (detailed comments below). The experiments also do not correspond well to the claims made in the introduction and abstract. This paper is also very hard to read which makes understanding the proposed method and other details quite challenging.

Novelty: The novelty of this paper mainly stems from its proposed method of using prototypes which serve as positive and negative examples w.r.t. a specific concept, and leveraging the weight vectors obtained when predicting the positive/negative classes using activation layer outputs to understand the influence of concepts of interest.  The technical novelty of the proposed approach is fairly minimal. The experiments also do not support a lot of novelty claims made about the proposed approach.

Other detailed comments:
1. I would first encourage the authors to improve the overall presentation and organization of this paper.
2. Please add some intuition about the approach in the introduction. Also, please be succinct in explaining what kind of interpretability is provided by the explanations. I would advise the authors to refrain from making very broad claims and using words such as hypothesis testing without discussing them in detail later in the paper.
3. Sections 2.3 and 2.4 are quite confusing and can probably be organized and titled differently. In fact, I would advise the authors to structure related work as i. inherently interpretable models ii. global explanations
iii. local explanations iv. neuron level investigation methods. Highlight how existing methods do not incorporate plug-in readiness and/or other desiderate wherever appropriate within these subsections.
4. Additional related work on inherently interpretable models and global explanations:
i. Interpretable classifiers using rules and Bayesian analysis, Annals of Applied Statistics, 2015
ii. Interpretable Decision Sets: A joint framework for description and prediction, KDD, 2016
iii. A Bayesian Framework for Learning Rule Sets for Interpretable Classification, JMLR, 2017
iv. Interpretable and Explorable Explanations of Black Box Models, FAT ML, 2017
5. In section 3, clearly identify what are the inputs and outputs of your method. Also, clearly highlight the various ways in which outputs of your method can be used to understand the model behavior. While Secction 3.2 and 3.3 attempt to describe how the CAV can be used to explain the model behavior, the presentation in these sections can be improved.
6. I think the experimental sections suffers from the following shortcomings: i. it does not substantiate all the claims made in the introduction ii. some of the details about which layer outputs are being studied are missing through out the section.

Overall, while this paper proposes some interesting ideas, I think it can be improved significantly in terms of its clarity, claims, and evaluation.

---

### Official Review · AnonReviewer5 · 2017-12-13
**Interesting Ideas But Needs Better Exposition and Validation**

**Rating:** 5
**Confidence:** 2

**Review:**

The paper deals with concept activation vectors, which the authors aim at using for interpretability in deep feed-forward networks. This is a critical sub-field of deep learning and its importance is only rising. While deep networks have yielded grounbreaking results across several application domains, without explanations for why the network predicts a certain class for a data point, its applicability in sensitive fields, such as medicine, will be limited. The authors put forth four desiderata and aim to construct a methodology that satisfies all of them. The concept vector is the 2-class logistic regression solution that discriminates between two classes of images (a grounded idea and other). This vector is used to amplify or diminish the effect of a concept at a certain layer, thus leading to differing output probabilities. The difference in probability can be used to understand, qualitatively, the importance of the concept. I have a few major and minor concerns, which I detail below.

* The structure and exposition of the paper needs to be significantly improved. Important sections of the paper are difficult to parse, for instance, Sections 2.3 and 2.4 seem abrupt. Also, the text and the contributions have a mismatch. The authors make several strong claims (hypothesis testing, testable quantifying information, etc.) about their approach which are not entirely validated by the results. The authors should especially consider rewriting portions of Sections 1 and 2; many of the statements are difficult to understand. There are many instances (e.g., the ears of the cat example) where a picture or graphic of some kind will greatly benefit the reader. What would also be useful is a Table with the rows being the 4 desiderata and the columns being various previous approaches.

* Am I right in assuming that the concept vector discriminator is simple (un-regularized) logistic regression?

* I don't quite understand why the weights of a discriminator of activations stands as a concept activation vector. The weights of the discriminator would be multiplied by the activations to figure out whether are in the concept class or not; I especially don't grasp why adding those weights should help tease the effect.

* Is the idea limited to feed-forward networks, or is it also applicable for recurrent-like networks? If not, I would encourage the authors to clarify in the title and abstract that this is the case.

* For Equation (1), what is the index 'i' over?

* In reference to Figure 1, have you experimented with using more data for the concepts that are difficult to discriminate? Instead of asking the practitioners for a set amount of examples, one could instead ask them for as much as to discriminate the classes with a threshold (say, 70%) accuracy.

* In the same vein, if a certain concept has really poor predictability, I would assume that the interpretability scores will be hampered as well. How should this be addressed?

* The authors desire a quantitative and testable explanation. I'm not sure what the authors do for the latter.

---

### Official Review · AnonReviewer4 · 2017-12-13
**Naive approach with minimal novelty. Need theoretical proof with thorough evaluations**

**Rating:** 3
**Confidence:** 5

**Review:**

This paper tries to analyze the interpretability of a trained neural network, by representing the concepts, as their hidden features (vectors) learned on training data. They used images of several example of a concept or object to compute the mean vector, which represent the concept, and analyzed, both qualitatively and quantitatively, the relationship between different concepts. The author claimed that this approach is independent of concept represented in training data, and can be expanded to any concepts, i.e. zero shot examples.

Major comments:

1- The analysis in the experiment is limited on few examples on how different concept in the training set is related, measures by relative importance, or not related by created a negative concept vector of un related or random images. However, this analysis severely lacks in situation where training set is limited and induces biases towards existing concepts

2-The author claims that this approach encompass following properties,
accessibility: Requires little to no user expertise in machine learning.
customization: Adapt to any concept of interest (e.g., gender) on the fly without pre-listing a set of concepts before training.
plug-in readiness: Work without retraining or modifying the model.
quantification: Provide quantitative and testable information.

Regarding 1) analyzing the relationship between concepts vectors and their effect of class probability need some minimal domain knowledge, therefore this claim should be mitigated
Regarding 2) Although some experiment demonstrates the relationship between different colors or properties of the object wearing a bikini can shed a light in fairness of the model, it is still unclear that how this approach can indicates the biases of training data that is learned in the model. In case of limited train data, the model is incapable of generalize well in capturing the relationship between all general concepts that does not exist in the training data. Therefore, a more rigorous analysis is required.
Regarding 3) compare to deepdream that involved an optimization step to find the image maximizing a neuron activation, this is correct. However, guided back propagation or grad-cam method also does not need any retraining or model tweaking.

Minor comments:

1- there are many generic and implicit statements with no details in the paper which need more clarification, for example,

Page 4, paragraph 2: “For example, since the importance of features only needs to be truthful in the vicinity of the data point of interest, there is no guarantee that the method will not generate two completely conflicting explanations.”

2- equation 1: subscript “i” is missing

3- section 4.2: definition for I^{up/down} of equation 1 is inconsistent with the one presented in this section

---

> ### Author Response · Authors · 2017-12-15
> **Author's response**
>
> Reviewer Comment:“1) analyzing the relationship between concepts vectors and their effect of class probability need some minimal domain knowledge, therefore this claim should be mitigated”
>
> Author’s response: We believe that any users using TCAV will have some representative domain knowledge about the data they are working with; we did not intend to indicate otherwise. However, explanations should preferably not require any understanding of the decision-procedure mechanisms or Machine Learning.  As our experiments show, TCAV can meet this bar by highlighting relationships and correlations of user-defined concepts to the model’s prediction.
>
> (To be clear: We think if the user does not even have minimal domain knowledge, any explanation method would fail. For example, if someone is trying to explain how a biokinematics decision procedure works, the audience has to understand the basics of biokinematics to assess the explanatory data being presented, and how it is correlated to decisions.)
>
> Reviewer Comment:“ 2) Although some experiment demonstrates the relationship between different colors or properties of the object wearing a bikini can shed a light in fairness of the model, it is still unclear that how this approach can indicates the biases of training data that is learned in the model. In case of limited train data, the model is incapable of generalize well in capturing the relationship between all general concepts that does not exist in the training data. Therefore, a more rigorous analysis is required.”
>
> Author’s response: Could you please articulate? It seems to us that this comment indicates that the results in this paper does shed a light in exposing biases, but you have particular analysis in mind that would further strengthen the paper. We would love to hear it.
>
> Reviewer Comment:“3) compare to deepdream that involved an optimization step to find the image maximizing a neuron activation, this is correct. However, guided back propagation or grad-cam method also does not need any retraining or model tweaking.”
>
> Author’s response: We mention in section 2.1 that the saliency map methods do satisfy criteria 3). In fact, deep dream also satisfies 3), as noted in our section 2.2; the deep dream technique does not require retraining the model. The argued advantage of TCAV is that it simultaneously satisfies all 4 desiderata whereas previous techniques do not.

---

### Public Comment · (anonymous) · 2017-12-12
**comparison to network dissection paper**

It seems that the proposed approach is similar to recent Network Dissection paper by David Bau et al. presented at CVPR 2017. Did authors think about comparison?

---

> ### Author Response · Authors · 2017-12-14
> **Author’s response**
>
> As we reference this work (ND paper) in our paper (Section 3.2), we also note that this work is a great step towards understanding NNs. Despite the similarities in searching for human-relatable concepts in layers, the two works are not directly comparable. The ND paper focuses on identifying individual units that detect a concept, while this work focuses on a direction in the entire layer that represents a concept. Finding directions in a layer is strictly more general than finding individual neurons as the learned CAV could in general be sparse. The ND paper improves the scientific understanding of CNNs (i.e., the relationship between interpretability and discriminative power in the paper’s Figure 4), whereas the goal of this work is to offer quantitative explanations of the relative importance of each concept via z-testing.

---

### Author Response · Authors · 2017-12-15
**Author's response**

Many thanks to the reviewers for their thoughtful and helpful comments. We are glad that the reviewers clearly saw the potential for this work on interpreting NNs. We have uploaded a new version of the paper which contains significant changes from the original version (please see the revision). With the help of the reviewers comments we have significantly improved the presentation and added a number of clarifying details throughout the paper.

We also added the details of how we apply hypothesis testing in order to obtain quantitative explanations. These details were not included in our earlier version. This addresses several of the reviewers' comments on the lack of evidence on testability. Given samples of class images (e.g., zebra pictures) and two concept vectors A and B, we perform two-tailed z-testing to invalidate the null hypothesis that there is no difference in importance of concepts A and B for the class. We perform this testing for each pair of concepts.

We address common concerns in this thread. We also individually addressed the comments from two reviewers in individual threads.

Reviewer Comment: “The technical novelty of the proposed approach is fairly minimal. The experiments also do not support a lot of novelty claims made about the proposed approach.“
Author’s response: Could you please provide more information regarding the lack of novelty? In the related works section we detail why previous methods do not meet all 4 desiderata that we describe in the introduction. Do you have a related work in mind that meets all of these desiderata? We view the simplicity as a strength of the work and not a weakness. We believe this work should be judged based on the novelty of the desiderata it satisfies over previous approaches and not by the technical complexity of the method.

---

> ### Author Response · Authors · 2017-12-15
> **Author's reponse**
>
> Reviewer Comment: “...But this paper provides no such metric. The explanations in figure 5 are not presented as good or bad according to any metric.”
>
> Author’s response: This is a great point and we will clarify the metric that we provide and experiments we conduct in order to detail the advantages the TCAV method has over saliency methods in terms of quantification.
>
> TCAV is quantitative and directly ties the quantification to the explanation given in a testable manner. For example, if I_w+^up is higher for red than yellow in relation to fire engine then the explanation is that the concept of red is more important to the classification of firetruck than the concept of yellow. The relative differences in the magnitudes of I_w+^up for red and yellow is thus the measure of the relative importances of the concepts. This is made more precise by testing against the null hypothesis that no color is significant. To test this hypothesis we can do two tailed z-testing on the measured importance values, and ask the question what is the probability that random concept vectors would observe the measured difference. The p-value of this test is thus the metric we use.
>
>
> By contrast, while saliency does produce a quantitative measure of “importance” for each pixel in the image, the user still needs to figure out how to use these quantities in order to interpret the network. Simply looking at the saliency map is a qualitative explanation, and prior work applying saliency methods to image datasets has used qualitative comparisons as part of the evaluation of their methods. For example [1, figure 2], is a qualitative comparison of the saliency maps produced by two different methods and the authors note that their method is better at identifying distinctive features in the image. [1, figure 3] also provides a qualitative explanation of which areas of the image are most important for classification. Although this qualitative explanation of the most important regions ultimately results from quantitative pixel intensities, it is unclear how the exact pixel intensities relate to the measure of “importance”. For example the authors in discussing figure 3 note that the saliency highlights the boundary of the area of interest with large positive values and the interior with large negative values. The authors interpret this as meaning the network focuses on the boundary of these regions and not the interior. However, it is still unclear how relative pixel intensities relate to importance. If one area of the image is brighter than the other, does this imply greater importance? How should brightness of a region be quantified? Should a bounding box be drawn and the mean saliency value in the region be computed?
>
>
> [1] - “Axiomatic Attribution for Deep Networks”, Sundararajan et. al.
>
> Reviewer Comment:“How practical is this method? Is it a significant burden for users to provide 100-500 images per concept? Are the top 100 or so images from a search engine good enough to specify a CAV?”
>
> Author’s response: We acknowledge that this method does require some effort from the users to curate a dataset.
> For example, we believe taking the top 100 images from a search engine with some manual curation to remove irrelevant results is sufficient. In fact, the “arms” concept presented in the paper is curated in that way, using only 33 images. We believe this extra effort is well worth the customizability that the TCAV method provides. In our experience, end-users always have different concerns/hypothesis in mind that they are eager to test based on their domain expertise. If the end-user does not have a particular hypothesis in mind, a simple way to start is to collect a set of data points with the same feature (e.g., in case of categorical data, collect all data points that has the same feature values for a set of features of interest), and conduct hypothesis testing on each set as a concept.
>
> Reviewer Comment:“The second use of CAVs, to test relative importance of concepts, is basically an improved saliency method. It's advantages over other saliency methods are stated clearly in 2.1, but it should not be portrayed as fundamentally different.”
>
> Author’s response: Thank you for pointing out the relationship between saliency methods and this work, it is an important connection to make and we will rewrite 2.1 to make this connection clear. Also thank you for the link to directional derivatives, we will also specify this connection and the updated paper will contain experiments which computes I_w^up formally as a directional derivative.
>
>
> Reviewer Comment:“Maybe consider some evaluation options from section 3 of Doshi-Velez and Kim 2017 (cited in the paper).”
>
> Author’s response: Great point, we plan to add a small scale study in the final version of the paper.

---

> > ### Comment · AnonReviewer1 · 2017-12-15
> > **Please provide more details**
> >
> > Thanks for the re-write. This has improved the paper, though it has made it 12 pages long and I still have some concerns.
> >
> > Author Response: "TCAV is quantitative and directly ties the quantification to the explanation... The p-value of this test is thus the metric we use."
> >
> > I can not find these p-values in the paper. Ideally, for every selected class k these should be reported for all all concepts C (e.g., instead of just using "yellow, green, blue, red" for the Fire engine class). Additionally, at least some classes should be selected randomly and the method for choosing random concept vectors should be described.
> >
> > The proposed experiments only establish that CAVs have a significant effect on model outputs. Showing this will be enough for me to increase my rating, but I'm still more interested in what effect TCAV has on human understanding of models. I would increase my rating further if this kind of evaluation was provided.
> >
> > Here's a suggested experiment along those lines: Take the top 5 classes the model predicted for a certain example. Provide the subject who is trying to predict model behavior with CAV explanations for each of these classes and the image. Ask the user to predict which class the model actually ranked 1st (and maybe which the model ranked 2-5 as well). If humans with TCAV do better than humans without TCAV then I'm much more comfortable saying TCAV helps with interpretability.
> >
> >
> > Author response: "Thank you for pointing out the relationship between saliency methods and this work, it is..."
> >
> > The incorporation of directional derivatives wasn't quite correct: "saliency maps take the derivative of the logits with respect to each pixel, while our work takes derivatives with respect to a concept direction." It should be that TCAV takes derivatives "in the direction of" a concept, not "with respect to" a concept. Directional derivatives are not gradients.
> >
> > I think the paper's presentation of saliency techniques is a bit misleading.
> > There is a significant difference between TCAV and saliency methods (criteria 2), but I don't think either of these preclude the kind of quantification (criteria 4) described in this paper.
> > You can measure how an image before/after DeepDream changes class probablility.
> >
> >
> > I'd also like to reiterate some concerns, mainly about missing details:
> >
> > * Relative vs absolute importance and user choice: Section 4.2 claims that figure 5 shows that a CAV "captures an important aspect of the prediction." I would be a bit more careful about the distinction between relative and absolute here. If red makes images more probably fire trucks then it doesn't necessarily mean that red is important for the fire truck concept in an absolute sense. Can we be sure that there aren't other concepts which more dramatically affect outputs? What if a user makes a mistake and only requests explanations with respect to concepts that are irrelevant to the class being explained? Do we need to instruct users on how to best interpret the explanation?
> >
> > * TCAV measures importance/influence as an average over a dataset. This is a validation set, right? For how many of these images are both the user concept and target concept unrelated to the image content (e.g., stripes and zebra for an image of a truck)? When that happens is it reasonable to expect meaningful explanations? They may not be meaningful because the data distribution used to train the CAV probably does not even sparsely cover all concepts in the network's train set. (related to "reference points" in "The (Un)reliability of Saliency Methods" submitted to ICLR18)
> >
> > * For relative importance testing it would be nice to see a note about the step size selection (1.0) and experiments that show the effect of different step sizes. Hopefully influence is monotonic in step size so that different step sizes do not significantly change the results.
> >
> > * How large is the typical difference between p_k(y) and p_k(y_w) in eq. 1? If this difference is small then is it meaningful? Are small differences signal or noise?

---

### Decision · Program_Chairs · 2018-01-29
**ICLR 2018 Conference Acceptance Decision**

**Decision:**

Reject

**Comment:**

This paper does not meet the acceptance bar this year, and thus I must recommend it for rejection.